# Effect of Drought Stress on Quality of Flax Fibres

**DOI:** 10.3390/ma16103752

**Published:** 2023-05-15

**Authors:** Edyta Kwiatkowska, Małgorzata Zimniewska, Patrycja Przybylska, Barbara Romanowska

**Affiliations:** Department of Innovative Textile Technologies, Institute of Natural Fibres and Medicinal Plants National Research Institute, 60-630 Poznan, Poland

**Keywords:** drought stress, flax, fibre

## Abstract

Global warming has led to a change in climatic conditions. Since 2006, drought has contributed to a reduction of food production and other agriculture-based products in many countries. The accumulation of greenhouse gases in the atmosphere has caused some changes in the composition of fruits and vegetables, making them less nutritious. To analyse this situation, a study was conducted on the effect of drought on the quality of fibres provided by the main fibre crops in Europe, namely flax (*Linum usitatissimum*). The experiment consisted of growing flax under controlled comparative conditions with designed different irrigation levels, such as 25%, 35% and 45% field soil moisture. Three varieties of flax were grown in 2019, 2020 and 2021 in the greenhouses of the Institute of Natural Fibres and Medicinal Plants in Poland. Fibre parameters, such as linear density, length and strength, were evaluated according to relevant standards. In addition, scanning electron microscope images of the cross-section and longitudinal view of the fibres were analysed. The results of the study indicated that deficiency of water during the flax growing season resulted in lowering of fibre linear density and tenacity.

## 1. Introduction

The world, including Europe, is struggling with the consequences of progressing global warming (Figure 1). The phenomenon of global warming, an increase in the temperature of the Earth’s surface, is occurring mainly due to the greenhouse effect, which results from the increased amount of greenhouse gases in the atmosphere [1]. Changes in the concentration of greenhouse gases, which include 90% carbon dioxide, methane and nitrous oxide, are mainly caused by certain human activities, including:The burning of fossil fuels, which is responsible for about 83% of the increase in CO_2_ in the air over the past 20 years;Land exploitation, mainly deforestation, which is the second largest factor after fossil fuels and responsible for the increase in carbon dioxide in the air;Emissions of methane, which are produced by agricultural processes, mainly from animal breeding.

**Figure 1 materials-16-03752-f001:**
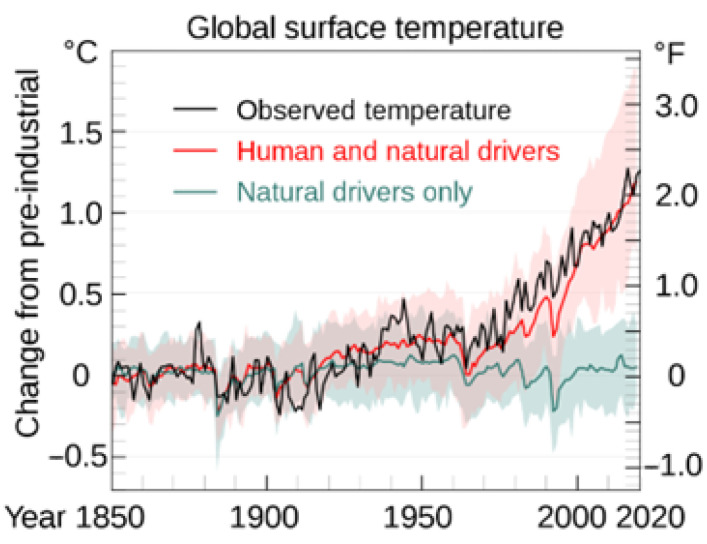
Increase of global surface temperature caused by human and natural drivers from 1850 to 2020 [2].

In addition to human activities, volcanic eruptions can have very large negative contributions to the degradation of the Earth’s climate; nevertheless, carbon dioxide emissions from volcanoes account for 1% of human CO_2_ emissions into the atmosphere.

Global warming is leading to extreme weather phenomena, such as heat waves, droughts and intense storms. Böhnisch et al. [3], in their study, indicated that weather anomalies will occur more frequently and their intensity will be increased. Higher air temperatures will cause more intense evaporation and drying of the soil [3]. The frequency of summer droughts in Europe is expected to increase sharply, by an average of about 25% for the entire region, and in the case of France by as much as 60% by 2100 [4]. Around 25–30% of CO_2_ produced by human activity, such as burning fossil fuels for energy, transportation, industrial processes, agriculture and livestock or deforestation, is absorbed by land ecosystems, which has an effect on temperature extremes and anomalies caused by the evaporation of water from soils. The interaction between soil moisture and vegetation is conditioned by climate systems, including the occurrence of such phenomena as droughts and heatwaves [5]. A lack of water during the growing period results in limited plant development, which leads to the production of smaller plant components, hampered flower production and grain filling [6]. Among abiotic stresses, defined as the negative impact of non-living factors on living organisms in a specific environment, drought is one of the most important yield-limiting factors. Other stresses include salinity, low or high temperatures, and other environmental extremes [7,8].

Under water-deficient conditions, cell elongation in tall plants is inhibited by reduced tissue tension. In the same way, drought stress reduces photo-assimilation and the metabolites necessary for cell division. The consequence is impaired mitosis and inhibition of cell growth and development [6]. A schematic mechanism of plant growth reduction under drought stress is presented in Figure 2.

Exposure of plants to abiotic stresses, including drought stress, leads to the formation of reactive oxygen forms (ROS), such as singlet oxygen (O_2_), perhydroxyl radical (HO_2_), hydroxyl radicals (HO), hydrogen peroxide (H_2_O_2_) and alkoxy radical (RO), which can react with proteins, lipids and DNA, causing oxidative damage and impairing normal cellular functions [6]. The increased stress of drought due to progressive climate change and awareness of the destructive effects of water scarcity on plant development led the authors to undertake this work, aiming to study the effects of drought stress on the fibre quality parameters of flax. Flax is a plant of great importance for the development of a sustainable bioeconomy, providing raw materials for the production of bio-products in the textile, composite, construction, food and medical sectors.

## 2. Materials and Methods

### The Schedule of the Experiments

The subjects of the study, flax (*Linum usitatissimum)* of the varieties Artemida, Modran and Sara, were grown under controlled conditions, including maintaining constant soil moisture at 25%, 35% and 45% of field soil water capacity. Climatic conditions and soil composition were the same for all plants in a given year. The experiment was conducted in three consecutive years, i.e., 2019, 2020 and 2021. The experiment is illustrated in Figure 3.

The experiment was set up in metal pots of equal diameter (30 cm) in 12 replicates for each flax variety tested in a vegetation hall at the Experimental Farm of the Institute of Natural Fibres and Medicinal Plants National Research Institute in Petkowo. The pots were tared and filled with a constant mass of soil taken from the field applied with the correct crop rotation for flax cultivation. The soil was tested for macro- and micronutrient content, and humus and granulometric composition were determined (Table 1) using relevant standards.

The growth and development of flax was evaluated with the BBCH scale (Biologische Bundesanstalt, Bundessortenamt und Chemische Industrie). This is a unified system and precise tool for assessing the state and degree of growth and development of crop plants. It enables farmers to carry out treatments at optimal agrotechnical periods. Linnaean plants were assigned to 9 main developmental stages (Table 2) [20].

Seeds were sown in the spots—50 seeds per pot, at a depth of 1 cm. After sowing, the soil surfaces in each pot were covered with a layer of quartz sand (300 g with a thickness of 0.5 cm) to protect against excessive drying and soil crusting. After sowing the seeds until the start of the rapid growth phase, i.e., when the height of the flax plants reached 18–20 cm (according to BBCH 32), the soil moisture in the pots was maintained at the optimal level, i.e., at 45% field water capacity (FWC), which refers to the maximum amount of water that soil can hold after it has been saturated and excess water has drained away. This is the amount of water that remains in the soil and is available for plants to use. The field water capacity is determined by the texture, structure and organic matter content of the soil, as well as the depth of the soil and the amount of rainfall or irrigation. Soils with high field capacity retain more water and require less frequent irrigation, while soils with low field capacity dry out more quickly and may require more frequent irrigation to support plant growth.

After the equalization of emergence, when the plants reached an even and harmonious level in cultivation, especially at the stage of germination and early growth, when plants reached 5–6 cm in height (according to BBCH 11), the removal of flax plants in pots was carried out to the number of 30–32 pcs per pot, leaving healthy and equalized plants. When the plants reached the specified height of 18–20 cm (according to BBCH 32), i.e., the initial stage of rapid growth, a controlled soil water deficit level of 25% FWC (in 4 pots for flax) and 35% FWC (in 4 pots for flax) was introduced until the flax plants were harvested. A level of 45% FWC was also maintained as a control (in 4 pots for flax). The list of symbols of tested flax varieties grown under specific soil moisture conditions are presented in Table 3. The water was supplemented using the weighted method based on the determination of soil moisture and field water capacity, which was determined via the Wanschaty method. The experiment was conducted without the use of pesticides.

The flax was harvested at the green-yellow stage of fibre maturity (BBCH 83). The process of natural flax straw retting in the field was conducted for about four weeks, until well-retted straws were obtained, according to an experienced expert, maintaining all the required steps performed in this type of retting. The method of retting flax straw was intended to ensure the high bioactivity of the fibres in accordance with the results of the described studies [21].

## 3. Results and Discussion

### 3.1. Assessment of Soil Parameters

An unfavourable environment (e.g., lack of water, extreme temperatures, air pollution), nutrient deficiency and suboptimal soil acidity reduce the size and quality of flax yield. Flax has a short growing season; the plants show resistance to some pests and diseases, and can be grown with the use of a limited quantity of insecticides and fungicides [22]. Each crop imposes unique requirements for soil properties, fertility status and nutrient content in the right proportions.

Plants have limited ability to selectively uptake mineral nutrients from the soil. There are 17 elements that are most important for plant growth, yield and development. Elements are divided into macronutrients and micronutrients. Macronutrients mainly perform building functions in plants. Micronutrients, on the other hand, participate in many metabolic reactions and perform physiological functions in plants [23].

Iron is a major component of reductase as well as of many enzymes involved in photosynthesis and N_2_ fixation, among others, and an activator in the synthesis of chlorophyll and some proteins.Manganese is an essential component of enzymes involved in decarboxylation, hydrolysis and oxidation reactions; it is also involved in photosynthesis in the photolysis of water, and activates many enzymes involved in the metabolism of proteins, sugars and lipids.Boron participates in the formation of cell wall structures.Copper is an element which takes part in nitrogen management in plants and participates in numerous oxidoreductive reactions, i.e., in photosynthesis process and respiration.Zinc is a component of the following enzymes: carbonic anhydrase, carboxypeptidase and alcohol dehydrogenase, among others. It participates in the regulation of glucose metabolism and in protein synthesis [23].

The optimal level of pH for flax growth is app. 5.0–5.5. An increase in pH value causes an imbalance of macro- and micronutrients. Zinc deficiency can cause physiological depression of plants, leads to disruption of enzyme activity, inhibition of photosynthesis, production of reactive oxygen species and increased iron accumulation.

Some varieties of flax differ in their tolerance to elevated soil pH value and zinc deficiency. Growing such varieties can significantly reduce yield losses. Zinc deficiency and elevated pH levels can cause moderate inhibition of plant growth, leaf browning and yellow necrotic spots on leaves [24].

After obtaining the results of the soil composition tests, no additional fertilizers were used in order to avoid the risk of soil overfertilization.

The content of assimilable as well as non-assimilable nutrients in the soil varied from one year to the next of the experiment, but nevertheless corresponded to the values recommended for growing flax.

Nutrient uptake, by fibrous flax, during the first growing season—from the beginning of emergence (BBCH 10) to the plant reaching 12 cm height (BBCH 14–16)—is low. This is due to a poorly developed root system and slow plant growth [25].

This period is followed by the rapid growth phase (BBCH 32—BBCH 36–37) when daily plant growth averages 2–4 cm. From the beginning of the rapid growth phase (BBCH 32—plant height 20 cm), intensive nutrient uptake is observed, which is highest during the flower bud formation (BBCH 51–59) and flax flowering phase (BBCH 61–69). In turn, sufficient water volume during the period of intensive flax growth (BBCH 32–35—plant height 20–50 cm) determines the straw yield and the fibre content and quality of the straw yield [20]. Although plants need micronutrients in small amounts, they are, in certain quantities and proportions, alongside macronutrients, essential for proper plant growth and development. Deficiency of a particular nutrient causes disruptions in the plant’s metabolism, leading in the first instance to the inhibition of basic biological processes, including photosynthesis, a change in the distribution pattern of assimilates and to a greater susceptibility of plants to adverse environmental conditions, and consequently to a reduction in yield and deterioration in its quality [23].

### 3.2. Assessment of Fibre Parameters

The yield of flax fibres (Figure 4) was determined by the percentage of the total weight of flax straw. After harvesting and dew retting, the flax straws were weighed and mechanically processed to separate the fibres from the rest of the plant material. Fibre extraction from the retted straw, in order to obtain long fibre, was carried out using a laboratory breaking and scutching device (Czech Flax Machinery, Meřín, Czech Republic) at the Institute of Natural Fibres and Medicinal Plants—National Research Institute. The fibres were weighed to determine the total weight of usable fibres. Flax plants require adequate soil moisture during their growth period to produce healthy and high-quality fibres. The moderate increases in soil moisture increase the yield of flax fibres.

The linear density of the fibres was tested in accordance with PN-EN ISO 1973:2011 [26]. Fibres extracted from plants that grew under the greatest drought stress (25% field water capacity of the soil) demonstrated the lowest linear density (Figure 5). Comparing the flax varieties, the highest linear weight was found in the Sara variety, and the lowest in Artemida (Figure 6). This parameter increased with higher soil moisture content, reaching its highest value for the control moisture content (45% field water capacity of the soil). Fibres extracted from all tested flax varieties exhibited a similar pattern. In the stalk, the fibre is found in the form of glued bundles called technical fibre. Applied preliminary mechanical processes allowed us to split the technical fibre into smaller fibre complexes and elementary fibres. The results of the linear density tests indicated that the water deficit during the growing period probably had a negative effect on the achievement of adequate maturity of the fibre, thereby affecting its thickness (Figure 7).

Fibre length was tested in accordance with the standard BN-7511-16:1986 [27]. The amount of water supplied to plants during their growth determines the quality of fibre in the straw yield. Some plants escape drought by reducing growth, accompanied by a yield penalty [28]. It could be observed that higher soil moisture resulted in a fibre length increase (Figure 8), which is understandable, because optimal soil moisture (45%) ensured favourable conditions for plants and allowed them to achieve higher flax growth in contrast to growing in drought conditions.

The flax fibre mechanical properties, including tenacity (Figure 9), breaking force and elongation, were determined according to the relevant standard, PN-P-04676:1986 [29] (Table 4). In the years 2019 and 2021, the highest tenacity of flax fibre was characteristic for fibres that were extracted from plants growing under the greatest drought stress. In dehydration tolerance, plants potentiate to maintain metabolic activities at low tissue water potential [28]. In 2020, the described relationship for flax tenacity was not clearly confirmed because the highest fibre strength was recorded for the Modran variety grown under 45% PPW conditions, but the differences are within statistical error, meaning that the values of tenacity of fibres from cultivation in soil with 25%, 35% and 45% PPW should be considered to be at the same level.

When analysing the values of breaking force, in most cases, it should be emphasized that the analogy with the changes in the values of tenacity of the flax fibres tested meant that fibre extracted from plants grown at lower soil moisture were characterized by a higher breaking force.

In 2019, from lower soil moisture contents, weaker fibres were obtained but the differences in breaking force values were within statistical error. In terms of assessing the effect of the soil moisture in which the plant grew on the value of elongation of the fibre, it is difficult to clearly identify relationships. Variations in elongation values for individual plant varieties and the occurrence of drought stress were within the limits of error.

The significance of the specific strength results was tested via multivariate ANOVA. The comparison of individual variances resulting from the influence of a factor and the error variance showed that the variety of flax and the applied level of soil moisture (field capacity) had a significant impact on the result of the tenacity of flax fibres. There was no significant dependence between the tenacity of fibres from plants cultivated in the soil with different level of moisture (field water capacity) and a variety of flax. The Sara variety showed the highest statistical significance in terms of a relationship between soil moisture and tenacity compared to the other varieties (Figure 10). The relationship between fibre tenacity and moisture level of soil used for flax growing was linear and decreased with the increase of soil moisture (Figure 11).

SEM images of longitudinal views (Table 5) of the fibres were taken at ×250 magnification, at an accelerating voltage of 15 kV and at a table height of 20 mm. The images of flax show the characteristic features of bast fibres. In longitudinal view, the elementary fibre of flax has a cylindrical shape. The surface of the fibre is generally smooth, and sometimes longitudinal cracks and crevices can be observed on the surface. A characteristic feature is visible transverse and diagonal fractures, so-called offsets, knots and elbows. The applied water deficit during plant growth did not affect the appearance of cross sections and longitudinal views of the fibres. Nevertheless, SEM images of longitudinal views of the fibres confirmed the results of the linear density of the fibres. The fibres extracted from plants growing under lower soil moisture conditions were noticeably thinner than the control fibres. This resulted from the observed lower amount of pectin on the fibre surface, which caused better dividing of technical fibres on smaller fibre complexes. The control fibres were still glued by pectin into larger complexes, with more pectin visible on their surface, even though the fibres were retted at the same conditions.

## 4. Conclusions

Climate change is a current key environmental, societal and economic factor to be resolved by mankind, whether by governments, researchers and individuals. One of the main challenges related to climate change is counteracting greenhouse emissions and drought, which have a strong effect on human life, agriculture and bioeconomy. Progressive desertification of agricultural regions have forced researchers to look for solutions for plant growth in water-deficient conditions to ensure the delivery of high-quality agricultural raw materials. The results of our study conducted within this doctoral thesis proved that the shortage and insufficiency of water in flax vegetation period influenced the quality of fibre. Linear density of the flax fibre was lower when plants grew under stress conditions caused by drought. In the case of the mechanical properties of the tested varieties of flax fibres, superior mechanical properties were demonstrated in the fibres of plants cultivated in soil with lower moisture content. Flax requires sufficient water to produce fibres with optimal parameters for the textile industry: length, linear density and tenacity. SEM analysis showed that drought does not significantly affect the appearance and structure of cross-sections and longitudinal views of fibres. The flax cultivated in water-deficit conditions delivered fibres with lower linear density and better mechanical properties in comparison to plants cultivated in soil with optimal water content. It seems that drought during the growing season can improve flax fibre quality. Growing flax triggered self-preservation by strengthening fibres to survive in unfavourable conditions, but there is a high risk that only a small number of plants will reach maturity because drought significantly limits each agricultural crop.

## Figures and Tables

**Figure 2 materials-16-03752-f002:**
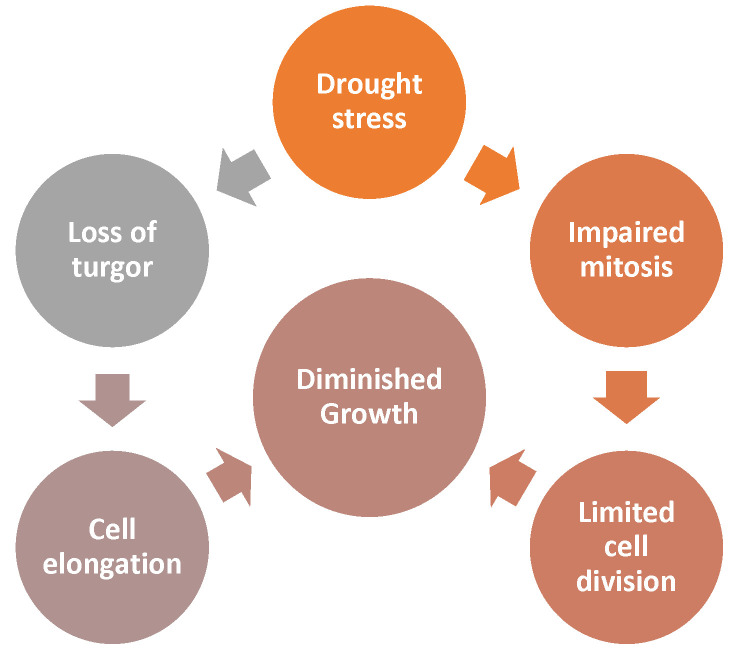
Schematic mechanism of plant growth reduction under drought stress, based on [6].

**Figure 3 materials-16-03752-f003:**
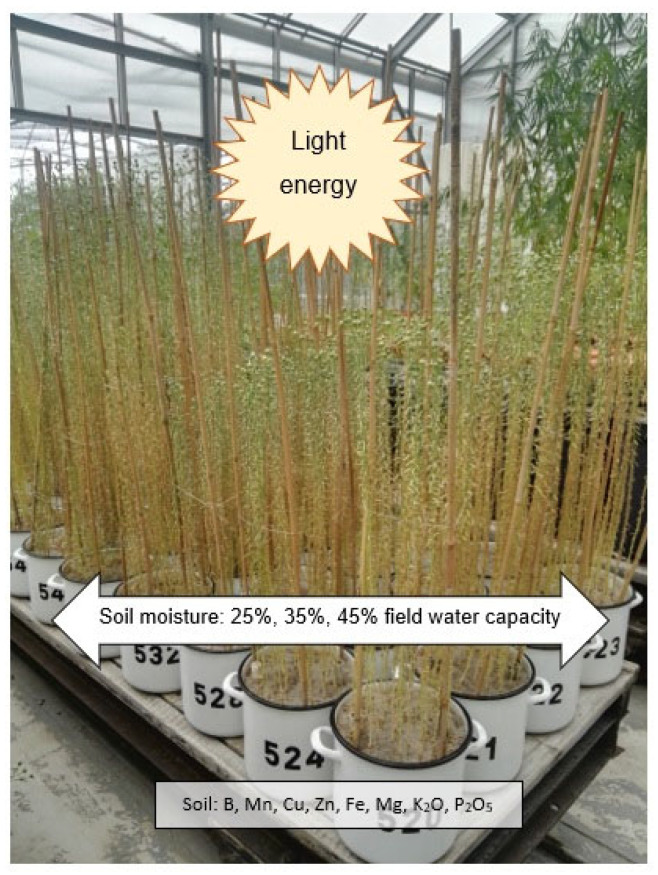
Model of experiment, flax cultivation in conditions of drought. Source: the authors’ materials, Edyta Kwiatkowska; Institute of Natural Fibres and Medicinal Plants—National Research Institute, Poland; 5 July 2019.

**Figure 4 materials-16-03752-f004:**
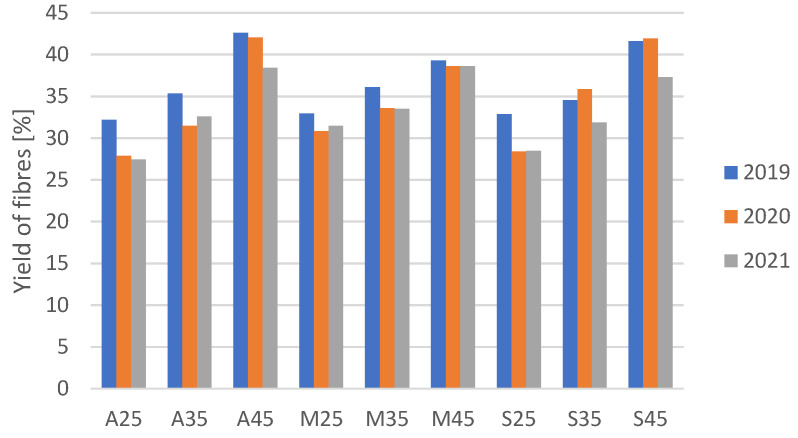
The yield of flax fibres from plants grown in soil with three moisture levels in 2019, 2020 and 2021.

**Figure 5 materials-16-03752-f005:**
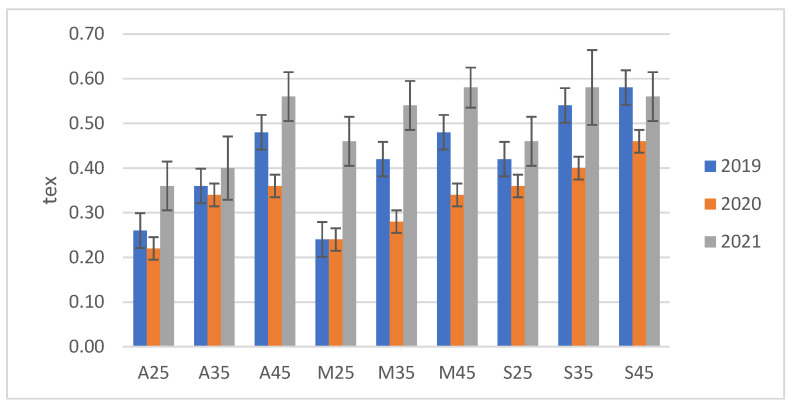
Linear density of flax fibres from plants grown in soil with three moisture levels in 2019, 2020 and 2021.

**Figure 6 materials-16-03752-f006:**
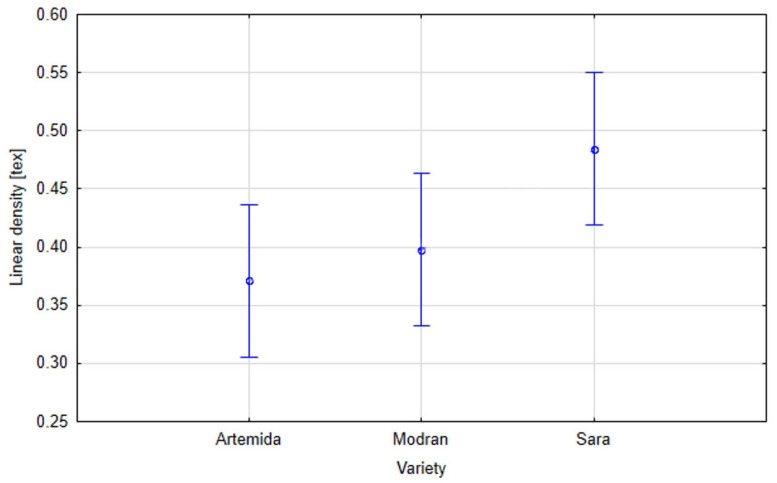
Effective hypothesis decomposition of flax variety and fibre linear density; *p* = 0.4914; vertical bars denote 0.95 confidence intervals.

**Figure 7 materials-16-03752-f007:**
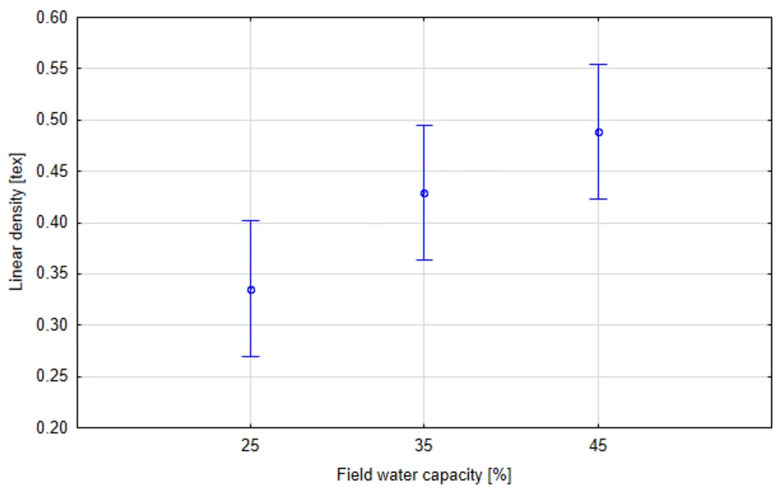
Effective hypothesis decomposition of soil moisture (field water capacity) and fibre linear density; *p* = 0.00957; vertical bars denote 0.95 confidence intervals.

**Figure 8 materials-16-03752-f008:**
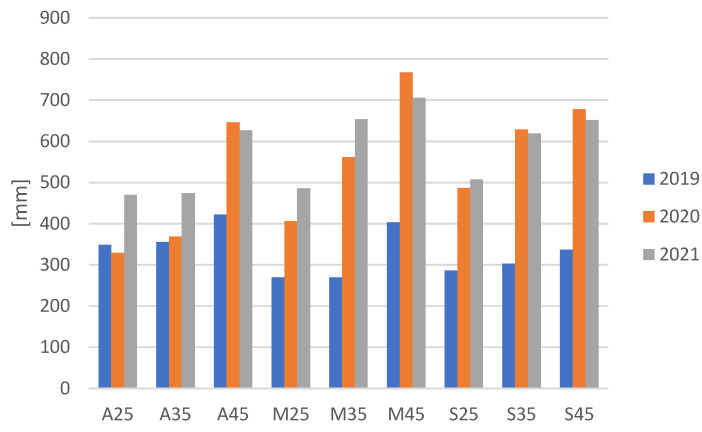
Average weight length of flax fibres from plants cultivated in soil with three moisture levels in the years 2019, 2020 and 2021.

**Figure 9 materials-16-03752-f009:**
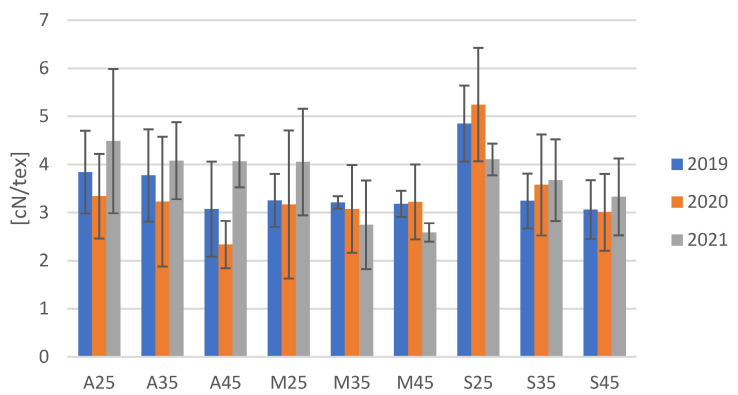
The tenacity of flax fibres derived from plants cultivated in soil with three moisture levels in the years 2019, 2020 and 2021.

**Figure 10 materials-16-03752-f010:**
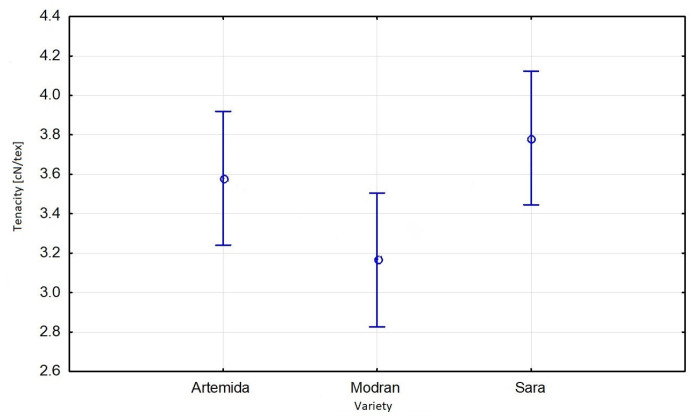
Effective hypothesis decomposition of flax’s variety and fibre tenacity; *p* = 0.4172; vertical bars denote 0.95 confidence intervals.

**Figure 11 materials-16-03752-f011:**
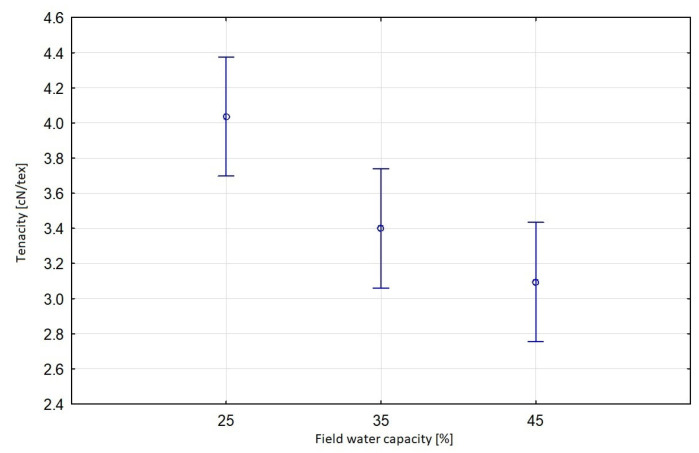
Effective hypothesis decomposition of soil moisture (field water capacity) and tenacity of flax fibres; *p* = 0.0021; vertical bars denote 0.95 confidence intervals.

**Table 1 materials-16-03752-t001:** Results of tests of the soil composition applied in pot experiment for flax growing in the years 2019, 2020 and 2021.

Conducted Test		Year		
		2019	2020	2021
Percentage content of mechanical fractions [9]	2.00–0.05	75.78%	71.22%	76.79%
	0.05–0.02	6.96%	9.35%	10.96%
	0.02–0.002	14.10%	16.31%	10.62%
	<0.002	3.17%	3.13%	1.64%
	Sands 2.0–0.05	75.78%	71.22%	76.79%
	Dusts 0.05–0.002	21.05%	25.66%	21.58%
	Loams <0.002	3.17%	3.13%	1.64%
	Mechanical composition	Loamy sand	Sandy clay	Loamy sand
Soil humus [10]		1.30%	1.92%	1.61%
Content of assimilable components [mg/kg soil] [11,12,13,14,15]	Boron B	0.55	1.39	1.15
	Manganese Mn	109.30	81.50	54.70
	Copper Cu	4.20	3.70	2.90
	Zinc Zn	7.90	10.10	6.70
	Iron Fe	805.00	534.00	423.00
Content of assimilable components [mg/100 g soil] [16,17,18]	P_2_O_5_	11.6	31.0	13.5
	K_2_O	11.7	27.1	18.2
	Magnesium MG	9.3	8.4	8.6
pH [19]		5.2	6.4	5.9

**Table 2 materials-16-03752-t002:** Main development stages on the BBCH scale for fibrous flax.

Flax Developmental Stages		BBCH Code from–to
Main development phase 0	Sprouting	00–09
Main development phase 1	Leaf development (main shoot)	10–19
Main development phase 3	Growth (elongation) of the main shoot	30–39
Main development phase 5	Inflorescence development	50–59
Main development phase 6	Blooming	60–69
Main development phase 7	Fruit development (green flax maturity)	71–79
Main development phase 8	Fruit and seed ripening	81–89
Main development phase 9	Aging, onset of resting period	97–99

**Table 3 materials-16-03752-t003:** List of symbols of tested flax varieties grown under specific soil moisture conditions.

Symbol	Variety of Flax	The Level of Drought Stress
A25	Artemida	25% field water capacity of the soil
A35	Artemida	35% field water capacity of the soil
A45	Artemida	45% field water capacity of the soil
M25	Modran	25% field water capacity of the soil
M35	Modran	35% field water capacity of the soil
M45	Modran	45% field water capacity of the soil
S25	Sara	25% field water capacity of the soil
S35	Sara	35% field water capacity of the soil
S45	Sara	45% field water capacity of the soil

**Table 4 materials-16-03752-t004:** Results of breaking force and elongation of flax fibres extracted from plants cultivated in soil with three moisture levels in the years 2019, 2020 and 2021.

Sample	Breaking Force [N]	SD [N]	Elongation [%]	SD [%]	Breaking Force [N]	SD [N]	Elongation [%]	SD [%]	Breaking Force [N]	SD [N]	Elongation [%]	SD [%]
A25	7.42	1.52	4.26	0.97	6.68	1.78	7.84	1.35	8.91	2.91	8.76	1.07
A35	7.62	0.96	5.00	0.82	6.15	2.47	8.54	1.96	7.78	1.63	7.87	0.62
A45	6.47	2.30	5.19	2.12	4.48	0.89	8.63	2.83	7.86	1.04	7.69	0.85
M25	6.57	1.12	6.40	1.96	6.05	3.01	10.40	5.68	8.21	2.29	9.06	0.45
M35	6.57	0.50	5.89	0.68	6.03	1.82	7.37	1.52	5.53	1.86	10.35	2.73
M45	6.36	0.62	5.71	0.85	6.36	1.30	7.96	1.43	5.28	0.45	7.89	0.67
S25	10.15	1.69	6.30	1.01	10.28	2.39	9.89	1.77	8.04	0.82	9.66	1.63
S35	6.53	1.14	4.35	1.63	7.03	2.12	9.96	4.43	7.15	1.78	8.15	0.61
S45	6.11	1.11	4.48	0.97	5.91	1.57	10.56	4.07	6.52	1.48	7.05	0.85

**Table 5 materials-16-03752-t005:** SEM images of longitudinal views of fibres.

Symbol	2019	2020	2021
A25	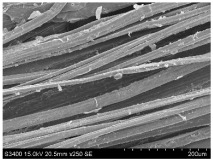	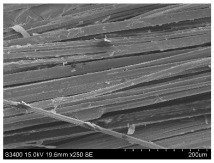	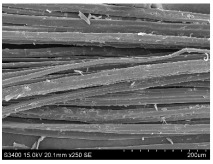
A35	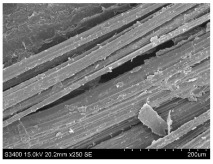	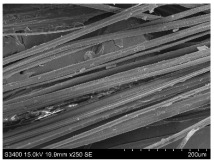	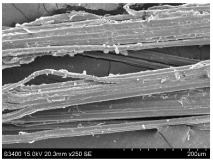
A45	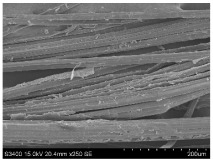	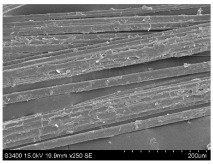	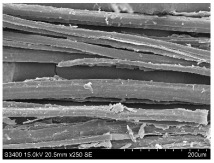
M25	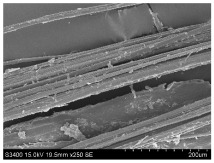	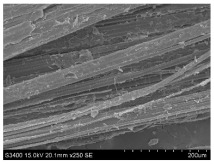	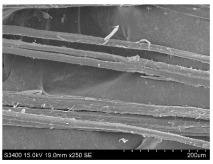
M35	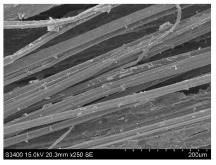	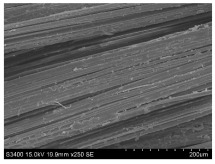	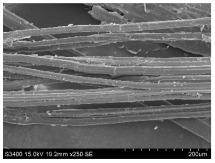
M45	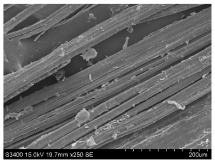	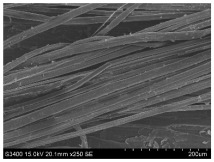	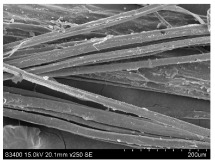
S25	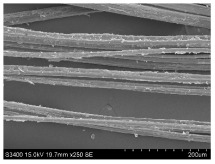	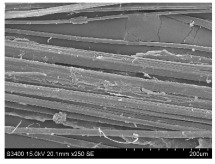	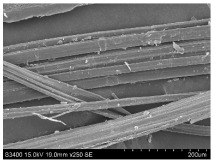
S35	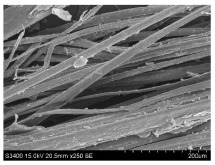	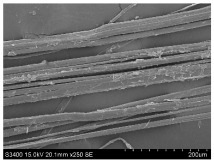	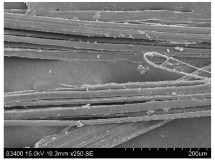
S45	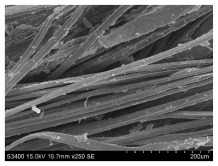	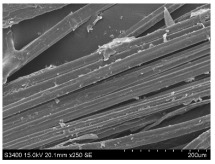	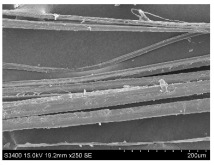

## Data Availability

Raw data are available upon request.

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
