# Peer review of "Effect of Drought Stress on Quality of Flax Fibres"

_materials, 2023, doi:10.3390/ma16103752_

Round 1

Reviewer 1 Report

The main thing seems to me which has to be done in section Results and discussion  is to integrate the cited resources into  discussion with own experimental  results. At the moment, it looks like some of what should be in the Introduction section has gone out of place. Normally, such tables as  Table 5 and 6 are not inserted into the article.  As for figure 8, it should be observed if there is a qualitative parameter on the horizontal axis (flax varieties), the deferred tenacity values with the line can not be connected. See the rest in the comments. 

Author Response

Dear Reviewer,

Thank you very much for all comments and suggestions on the article titled: Effect of drought stress on quality of flax fibres. Your comments are very valuable for me. In most cases, I took into account and correct them in the article. Below is the answer for the other one:

„Table 3: Soil composition applied in pot for flax growing in year 245 2019, 2020 and 2021 were rather different, and perhaps there were a reason for the change in the properties discussed below in parallel with the field water capacity of the soil.”

The aim of the study is to explore  drought stress on fibre properties, so study has beed focused on comparison of content of soil moisture influence on fiber properties. The composition of soil in each year was the same for each flax variety and for each field water capacity. This ensured appropriate results delivery to conduct relevant analysis for each year separately, because there was one variable e.g. field water capacity. The comparison of year to year of flax fiber properies was not aim of this study because FWC was the same for all years.

Yours faithfully,
Edyta Kwiatkowska

Author Response

Dear Reviewer,

Thank you very much for all comments and suggestions on the article titled: Effect of drought stress on quality of flax fibres. Your comments are very valuable for me. In most cases, I took into account and correct them in the article. Below are the answers for the other ones:

„The issue is not breeding of animals, but the number of animals and their type.”

Each animal breeding results in methane production. The amount of ethanol emission depends from the size of breeding herd.

„Need ANOVA on each of the key properties given 3 varieties, 3 years, 3 water levels. This will ensure that differences which are reported are significant. This will also mean each bar chart will have error bars (or at least)”

The length of the fibers was tested with use all fibes extracted from the whole flax straw growing during the experiment. Statistical analysis is not possible. ANOVA for linear density was added.

Yours faithfully,
Edyta Kwiatkowska

Round 2

Reviewer 1 Report

Thanks for taking note of the comments and thanks for the explanations

Author Response

Dear Reviewer,

thank you very much.

Yours faithfully,
Edyta Kwiatkowska